# In Vitro Evaluation of Antimicrobial Effect of Phytobiotics Mixture on *Salmonella* spp. Isolated from Chicken Broiler

**DOI:** 10.3390/antibiotics11070868

**Published:** 2022-06-28

**Authors:** Hubert Iwiński, Karolina Wódz, Karolina Chodkowska, Tomasz Nowak, Henryk Różański

**Affiliations:** 1Department of Chemistry, the Faculty of Food Science, Wrocław University of Environmental and Life Sciences, C.K. Norwida 25, 50-375 Wrocław, Poland; 2AdiFeed Sp. z o.o., Opaczewska, 02-201 Warszawa, Poland; k.chodkowska@jkrzyzanowski.pl (K.C.); rozanski@rozanski.ch (H.R.); 3Laboratory of Molecular Biology, Vet-Lab Brudzew, Turkowska 58c, 62-720 Brudzew, Poland; karolina.wodz@labbrudzew.pl (K.W.); tomasz@labbrudzew.pl (T.N.); 4Krzyżanowski Partners Spółka z o.o., Zakładowa 7, 26-670 Pionki, Poland; 5Laboratory of Industrial and Experimental Biology, Institute for Health and Economics, Carpathian State College in Krosno, Rynek 1, 38-400 Krosno, Poland

**Keywords:** *Salmonella* Typhimurium, *Salmonella* Enteritidis, phytobiotics, antimicrobial resistance, MIC, chicken broiler, resistance genes, thymol, 1,8-cineole

## Abstract

Background: The identification of natural antibacterial agents from various sources that can act effectively against disease-causing foodborne bacteria is one of the major concerns throughout the world. In the present study, a unique phytobiotics mixture containing thymol, menthol, linalool, *trans*-anethole, methyl salicylate, 1,8-cineole, and *p*-cymene was evaluated for antibacterial activity against selected strains of *Salmonella* spp. Results: The phytobiotics mixture was effective against *Salmonella enterica* subsp. *enterica* serovars Enteritidis, Typhimurium, and Kentucky. The minimum inhibitory concentration (MIC) and minimum bactericidal concentration (MBC) values of this unique mixture for these three pathogens were 1:256. Among these three strains, one *S*. Kentucky presented the most extensive resistance profiles to 18 antibiotics belonging to 5 classes of antibiotics. One of *S.* Typhimurium presents extensive resistance profiles to 14 antibiotics belonging to 5 classes of antibiotics. Conclusions: The results suggest that the phytobiotics mixture used in the experiment can be used as a strong natural antibacterial agent against Gram-negative foodborne pathogens such as *S.* Typhimurium, *S.* Kentucky, and *S.* Enteritidis. This is a preliminary analysis of the effectiveness of a phytobiotic product in an in vitro model, which may be the starting point for further studies, including in vivo animal models.

## 1. Introduction

Salmonellosis is the second most common zoonotic disease after campylobacteriosis in the European Union (EU), mostly related to eggs and raw meat from poultry production. Each year in Europe and North America there are 7–10 cases of foodborne illness (per 100,000 of the population) related to non-typhoid *Salmonella*. It causes 16 million cases of inflammatory fever including salmonellosis, 1.3 billion cases of gastroenteritis, and 3 million deaths from *Salmonella* worldwide each year. An increase in the confirmed human salmonellosis cases noticed in the last decade in the EU necessitated the search for a possible effective *Salmonella* control in broiler chicken flocks using various methods, including substances obtained from plants.

For several years, national *Salmonella* control programs (also in Poland) in broiler flocks have shown that the problem is not disappearing and is even becoming one of the main microbiological threats in poultry production. This, together with the increasing antimicrobial resistance of *Salmonella* spp. (mostly in humans), forced the necessity to undertake work on alternative methods of controlling and eradicating *Salmonella* in poultry production using promising antibiotic alternatives including probiotics, prebiotics, symbiotics, organic acids, essential oils, cinnamaldehyde, chitosan, phages, nanoparticles, and vaccines.

Antimicrobial resistance (AMR) is a global growing health threat in both human and veterinary medicine. Poultry production is one of the fastest and most dynamically developing animal productions in the world. This, together with the various conditions of poultry farming such as ineffective biosecurity, a high stocking density, or the increasing threat of viral diseases causing immunosuppression, significantly contributes to the increase in the use of antibiotics in veterinary medicine, and indirectly to the overall increased antibiotic resistance. The latest analysis showed that more than 70% of global antimicrobials produced on Earth are used in food-animal production [1]. Data about AMR transmission pathways and finding about how its spread in different part of the world, can be found in Table 1.

The antibiotics used in animals should be selected from those the World Health Organization (WHO) has listed as being “least important” to human health, and not from those classified and mentioned above as “highest priority critically important”. These antibiotics are often the last line, or one of limited treatments, available to treat serious bacterial infections in humans.

In 2019, WHO identified 32 antibiotics in clinical development that address the WHO list of priority pathogens. However, only six of them were classified as innovative. New antimicrobials agents, not only as classic antibiotics but also any alternatives (for, e.g., plant origin), are urgently needed in both human and veterinary medicine [3].

According to the Rapid Alert System for Food and Feed (RASFF) 2020 Annual report, the most incidents came from “accidental or environmental contamination” (30 incidents, 133 notifications), followed by “hazardous or unauthorized composition” (14 incidents, 42 notifications), and “foodborne outbreak” (9 incidents, 43 notifications). Among pathogenic microorganism groups related to the food sector analysed in the RASFF system, 788 notifications were related to—*Salmonella*, *Listeria monocytogenes*, *Norovirus*, and *E. coli*. The biggest number of *Salmonella* notifications (273) associated with poultry meat and poultry meat products was detected in Poland as a country of origin. Approximately half of these concerned *Salmonella* Enteritidis (149 notifications), for which a food safety criterion is set for fresh poultry. Sixteen operators were identified as recurrent. Number of notifications caused by *Salmonella,* associated to poultry meat and poultry meat products, divided to the by countries can be found in Table 2. These data clearly indicate that the problem of *Salmonella* in food of animal origin, in particular in poultry meat and its products, is still valid despite the introduction of many solutions both at the farm, slaughterhouse, and production plant level. [4].

The increasing resistance of microorganisms has generated the need to search for novel and, most importantly, effective solutions. As evidenced by current research, an effective alternative are substances produced by higher plants—phytoncides, phytoalexins, phytoanticipins, and other secondary metabolites. The term “phytoncide” was coined for the first time by the Soviet scientist B. P. Tokin [5]. This term describes substances secreted and excreted by telomeric plants that have antibacterial, antiprotozoal, and antifungal properties.

At the same time, scientists were also working on substances excreted by bacteria and fungi, trying to understand the mechanism of action and synthesize them. The milestone and first step to the “Golden Age” of antibiotics development was the discovery of penicillin by Alexander Fleming in 1928. On the other hand, the Second World War caused a huge demand for the substances with antimicrobial properties that can be easily used on the battlefield and prevent wounds infection. That reason forced the development of synthetic solutions over research on natural solutions. In the next 20 years, more than 25 antibiotics were discovered, and they dominated for the next decades [6,7]. Figure 1 shows the timeline and development of antibiotics in comparison to natural solutions.

The most recent term to describe plant secondary metabolites possessing antimicrobial properties is phytoanticipins. It was introduced by van Etten et al. and describes low-molecular-weight compounds characterized by antimicrobial properties, which are present in the plant before the appearance of the elicitor, while at the time of pathogen attack they are released or constitute the main component of their synthesis [8]. Such compounds include glucosinolates commonly found in Brassicaceae plants [9,10,11,12] or saponins exemplified by avenacins and a-tomatine [9,12].

Research is currently being conducted on the use of the antimicrobial properties of, i.a. alkaloids [13,14,15], saponins [16,17,18], flavonoids [19,20], and essential oils [21,22,23]. In this article we will focus on essential oils, more precisely on their constituents. Essential oils are volatile, poorly water-soluble, lipophilic natural substances. They are used in many industries such as food, cosmetics, medicine, and pharmaceutics [24]. Essential oils are natural volatile substances obtained from almost every part of plants, including the leaves, buds, flowers, bark, seeds, roots, rhizomes, stems, and fruits by steam distillation, supercritical fluid extraction, and solvent extraction [25,26,27].

Chemically they are mainly mixtures of terpenes (mono-, di-, and sesqui-), terpenoids, phenylpropanoids, isoprenoids, phthalides, and phenolic compounds [25,26,27,28,29].

Figure 2 presents some of the proposed potential mechanisms of antimicrobial action have been observed against bacteria in general, but it is suspected that they may be consistent with those observed for *Salmonella* spp.

The aim of the present study was to determine the antibacterial activity of phytoncides mixture (thymol, menthol, linalool, anethole, methyl salicylate, 1,8 cineole, and *p*-cymene) in vitro against *Salmonella* spp. isolated from broiler farms.

## 2. Results

The chemical composition of menthol, 1,8-cineole, thymol, *p*-cymene, anethole, linalool, and methyl salicylate was effective against *S.* Enteritidis, *S.* Typhimurium, and *S.* Kentucky. The minimum inhibitory concentration (MIC) and minimum bactericidal concentration (MBC) values of this unique mixture for all serotypes of these three pathogens were 1:256. In seven strains, those dilutions were even lower, reaching 1:512. Moreover, MBC for the liver *Salmonella* Enteritidis was 1:1024. Table 3. shows the dilution which was effective against analysed *Salmonella* spp. Appendix A show the evaluation of the MIC of the phytoncides mixtures on the analysed *Salmonella* spp. and the negative control.

### 2.1. Antimicrobial Resistance Profile

The isolates were subjected to antibiotic susceptibility tests against 33 antibiotics belonging to 10 different classes using the MIC method Merlin MICRONAUT (MERLIN Diagnostika GmbH, Niemcy, Germany) and the AST-GN 96 CARD and the VITEK2 system (Biomerieux, Marcy-l’Étoile, France). All *Salmonella* strains of the isolated species belong to 12 *Salmonella enterica* subsp. *enterica* and represented 3 serotypes (Typhimurium, Kentucky, Enteritidis). Antibiotic susceptibility testing conducted on the *Salmonella* strains showed that only two strains *S*. Typhimurium had resistance to two classes of antibiotics (CFX-CPH-GEN-NEO-STR) whereas other strains were resistant to three or more of the tested antibiotics. All isolated *Salmonella* were sensitive to imipenem (IMP) and colistin (COL)/polymyxin B (PB), cefequinome (CFQ), and trimethoprim/sulfamethoxazole (TR/SMX). Surprisingly, we detected that 100% of the *Salmonella* strains were phenotypically resistant to streptomycin and gentamycin. Amongst the resistant strains, only two resistance profiles were identical: CFX-CPH-GEN-NEO-STR (*S.* Typhimurium). *S.* Kentucky presented the most extensive resistance profiles to 18 antibiotics (AMP-AMX-AMX/CL-CFX-CFT-CPH-CFP-GEN-NEO-STR-ENR-UB-MRB-NOR-DOX-OXY-TET-LIN/SP) belonging to 5 classes of antibiotics. One of *S.* Typhimurium presents extensive resistance profiles to 14 antibiotics (AMP-AMX-CFX-CFT-CPH-CFT-CFP-GEN-STR-ENR-UB-MRB-FLR-LIN/SP) belonging to 5 classes of antibiotics. The classes to which it presented the highest resistance were β-lactams (AMP, AMX) and beta-lactam/beta-lactamase inhibitor combination (AMX/CL), I generation cefalexin (CFX-CFT-CPH), III generation cefalexin (CFT, CFP), aminoglycosides (GEN-NEO-STR), fluorochinolones (ENR-UB-MRB-NOR), and tetracyclines (DOX-OXY-TET). For individual serovars of *Salmonella* spp., Table 4 presents several multi antibiotic resistance patterns.

### 2.2. Prevalence of Multiple Drug Resistance

In our study, four of the isolates showed a multiple antibiotic resistance index (MAR index) greater than 0.3, whereas eight (all *S.* Enteritidis and two *S.* Typhimurium) showed a MAR index above 0.5. We observed a high prevalence of multiple antibiotic resistance amongst the isolates where four of the isolates were MDR, with resistance to five different classes of antibiotics.

### 2.3. Detection of Antibiotic Resistance Genes (ARGs)

Based on the antibiogram data, all isolates were studied for the presence of antibiotic resistance genes. The overall prevalence of ARGs amongst the investigated *Salmonella* isolates with their resistance phenotype is shown in Table 5. All the isolates were positive for at least one AMR gene. The gene *bla*_CMY2_, which confers resistance to ceftiofur (CFTI)/cefoperazone (CFP), was detected in four strains. However, the *Salmonella* Enteritidis strain did not exhibit phenotypic resistance to III generation cephalosporin. The genes *aad*A and *str*A/*str*B that confer resistance to streptomycin, were detected in all strains. The gene *aad*B was not detected in the strains. However, all *Salmonella* spp. strains were phenotypically resistant to gentamicin (GEN). All of the neomycin-resistant strains carried *aph*A1 and *aph*A2 genes. The *tet*A and *tet*B genes were detected in all strains resistant to doxycycline and oxytetracycline. The gene *flo*R that confers resistance to florfenicol was detected in all strains resistant to florfenicol. The distribution of the various resistance genes and the prevalence of the corresponding serovars are shown in Table 5.

## 3. Discussion

The results of the present study illustrated that the unique phytobiotic mixtures containing menthol, thymol, linalool, 1,8-cineole (eucalyptol), *p*-cymen, anethole, and methyl salicylate are effective against three *Salmonella* serotypes isolated from infected materials (birds, dust, and boot swabs) in an in vitro environment. To our best knowledge, there are no scientific reports showing bactericidal effects on *Salmonella* spp. of similar natural compounds composition.

The previous trend of studies related to searching for possible antibiotic alternatives (also used as antibiotic growth promoters) in animal production, so far has mostly focused on single components or mixtures of 2 to 3 active substances [30,31]. There are also reports with promising results, in the *Salmonella* spp. Control, with probiotics as an alternative for AGPs [32,33]. Analyzing the history of antibiotics from the discovery of penicillin in 1928 until today, when both human and veterinary medicine face the increasing drug resistance of many dangerous bacteria (including *Salmonella* spp.) and the lack of new, safe, and effective antibiotics, one should look at the dynamic development of phytobiotics. Phytobiotics, which include a wide range of plant-derived products such as essential oils, herbs, and other bioactive compounds, were first described in 1929 by Russian biochemist Boris Tokin who noticed that certain trees and plants release very active preservatives. This author conducted a number of studies in the 1950s, but only in the last 20 years have we observed a dynamic development of research on the potential effects of single phytobiotics or in simple combinations. Many recent studies show that the antimicrobial mechanism of action of phytobiotics is similar to the action of classic antibiotics [34,35], with the extremely important difference that in the case of phytobiotics, the phenomena of drug resistance and temporary accumulation of active substances in tissues (withdrawal period and antibiotic residues in products of origin) are not observed.

In our study, the sensitivity to 25 antibiotics were assessed. Penicillins (cloxacillin, penicillin G, and nafcillin), macrolides (erythromycin and tylvalosin), lincomycin, tiamulin, and tylvalosin were excluded from analysis, due to a natural lack of activity against *Salmonella*.

The results of the antibiotic resistance indicated that the *Salmonella* spp. strains isolated from birds could be categorized as resistant to MDR, that is, bacteria exhibiting resistance to one or more antibiotics from three or more classes of antibiotics. Phenotypic and genotypic profiles of resistance analysis showed that amongst these three strains, *S.* Kentucky presented the most extensive resistance profiles to 18 antibiotics belonging to 5 classes of antibiotics. One of *S.* Typhimurium presents extensive resistance profiles to 14 antibiotics belonging to 5 classes of antibiotics. These bacteria were resistant to β-lactams, aminoglycosides, cephalosporins, fluorochinolones, and tetracyclines. All strains were resistant to gentamicin, which is one of the major antibiotics used in the treatment of urinary infections in humans, and to streptomycin. Although streptomycin, an aminoglycoside, is not used for *Salmonella* treatment, streptomycin resistance has been widely used as an epidemiological marker [36]. For this reason, the results obtained in this study, indicating the antibacterial activity of the mixture of phytobiotics also against *Salmonella* Kentucky, seem to be extremely interesting and promising not only in terms of the control of salmonellosis in broiler breeding, but also due to the indirect reduction of resistance among this type of bacteria.

Finally, because these antibiotic phenotypes could be conferred by several ARGs, the detection of resistance genes was performed in order to confirm phenotypic antimicrobial resistance profiles.

In *Salmonella*, the main mechanism of resistance to β-lactams is encoded by the *bla* genes. The gene *bla*_CMY2_, encoding an extended-spectrum beta-lactamase that is responsible for resistance to ceftiofur (CFTI)/cefoperazone (CFP), was detected in four strains. However, the *Salmonella* Enteritidis strain did not exhibit phenotypic resistance to this antibiotic.

In our study, one *S*. Typhimurium and two *S*. Kentucky demonstrated the presence of the genes *bla*_PSE-1_ and *bla*_TEM_ that encode beta-lactamases and confer resistance to ampicillin. The genes *aad*A and *str*A/*str*B that confer resistance to streptomycin were detected in all strains. The gene *aad*B was not detected in the strains. However, all *Salmonella* spp. strains were phenotypically resistant to gentamicin. This resistance may be mediated by other resistance genes, which were not assessed in this study. 

The proposed composition shows unexpected and very high antibacterial properties against *Salmonella* spp. The composition was prepared based on experience of the authors with natural compounds and their uses in feed. All the components are allowed to be used as a feed additive according to Regulation (EC) no. 1831/2003 of the European Parliament and of The Council, Appendix I [37]. The composition based also on the scientific data provided by researchers about the used mentioned compounds against microorganisms. [38,39,40,41]. Similar trends have been reported by Kollanoor Johny et al. in their work on the plant-derived molecules trans-cinnamaldehyde, eugenol, carvacrol, and thymol [42]. However, their results test was conducted only on single compounds, not their mixtures and this might have been the reason for the lower efficacy of thymol. The synergistic effect of essential oils compositions was reported by Thanissery et al. The results show stronger antibacterial properties of a thyme–oregano blend against *Salmonella*, than both essential oils separately [43]. Mixtures with other natural compounds such as organic acids and surfactants present a very good effect against *Salmonella* spp. [44].

Our findings support the Aljumaah et al. hypothesis that supplementation of phytobiotic feed additives could be very effective in growth promotion, meat quality, and composition and what is most important, *S.* Typhimurium control [45].

The results obtained in this study, indicate that the antibacterial activity of the mixture of phytobiotics seems to be extremely interesting and promising not only in terms of the control of salmonellosis in broiler breeding, but also due to the indirect reduction of resistance among this type of bacteria.

## 4. Materials and Methods

### 4.1. Phytoncides Mixture

Seven common phytoncides were selected for the tests—thymol, menthol, linalool, *trans*-anethole, methyl salicylate, 1,8-cineole, and *p*-cymene. All compounds were purchased from Sigma-Aldrich (St. Louis, MO, USA) compliant to FCC and FG standards. Purity and percentage composition, according to supplier specification, was minimum ≥95%.

All phytoncides were mixed in equivalent amounts, heated, and left overnight. The prepared mixture was then mixed with an emulsifier (Polysorbat 80, Sigma-Aldrich) for easier dissolution in aqueous solutions and culture media.

### 4.2. Salmonella spp. Isolation and Identification

*Salmonella* spp. from environmental samples were isolated in accordance with PN-EN ISO 6579-1:2017-04. Microbiology of the food chain—horizontal method for the detection, enumeration, and serotyping of *Salmonella*—Part 1: Detection of *Salmonella* spp. Samples were taken from several infected broiler farms from different places (boot swabs and dust) and internal organs (intestines, liver, and spleen). Then, samples were suspended in 225 mL buffered peptone water (BPW GRASO, Starogard, Poland) in sterile stomacher bags for pre-enrichment (Whirl-Pak, NASco, Madison, WI, USA). The selective proliferation of *Salmonella* spp. was carried out using modified semisolid Rappaport–Vassiliadis (MSRV) agar (GRASO). Two selective enrichment media, xylose lysine deoxycholate agar (XLD, GRASO) and brilliant green agar (BGA, GRASO), were used as described [46]. *Salmonella* suspect colonies were transferred to nonselective nutrient agar (GRASO) to obtain the pure culture for testing flagellar antigens. Serotyping was performed by direct agglutination with commercial H poly antisera for verification of the genus *Salmonella* enterica (IBSS Biomed, Kraków, Poland), O group antisera to determine O group (IBSS Biomed, Kraków, Poland), and H phase and H factor antisera to determine H phase and H factor (IBSS Biomed, Kraków, Poland, Bio-Rad, Hercules, CA, USA), according to the White–Kauffmann–Le Minor scheme. Pure cultures were used for further biochemical and molecular tests.

### 4.3. Biochemical Strain Identification

Colonies showing morphology typical for *Salmonella* spp. on selective agars were subjected to biochemical identification using two commercially available tests: API 20E (BioMérieux, Craponne, France) and a VITEK2 COMPACT automated system for bacterial identification. VITEK^®^ 2 GN cards (BioMérieux, Craponne, France) with reference strains for *Salmonella* Typhimurium ATCC 14028, *Salmonella* Enteritidis ATCC 13076, and *Salmonella* Kentucky ATTC 9263 served as a quality control. Both tests were used according to the manufacturer’s instructions.

### 4.4. Confirmation of Salmonella Identification with Molecular Biology Methods

A real-Time PCR method based on the detection of genes specific for *Salmonella* spp. was used to confirm biochemical identification. DNA for Real-time PCR was extracted from bacterial cells using an automated method (AutoPure96, Wuxi, China). For detection of *Salmonella* spp., a commercial Kylt^®^ *Salmonella* spp. (Anicon, Emstek, Germany) kit was used. For simultaneous detection of *Salmonella* Enteritidis and *Salmonella* Typhimurium, a commercial Spp-Se-St PCR (BioChek, Reeuwijk, The Netherlands) kit was used. Both real-time PCR tests were performed according to the manufacturer’s instructions using an Applied Biosystems 7500 Fast Real-Time PCR System (Thermo, Waltham, MA, USA).

### 4.5. Phytoncides Mixture Test by Broth Microdilution Method

The antimicrobial activity of the phytoncides mixture was tested using the broth microdilution method described in ISO 20776-1:2006. In sterile vials, two-fold serial dilutions of the phytoncides mixture were prepared in Mueller Hinton II Broth (M-H Broth) with a final volume of 2 mL per vial. Next, fresh inoculum of each bacterial isolate was prepared by suspending colonies from an overnight culture on sheep blood agar in sterile saline (0.9% NaCl) and adjusting the turbidity 0.5 McFarland standard. Subsequently, the suspensions were diluted a hundredfold in M-H Broth by transferring 110 µL of the suspension into 11 mL M-H Broth. Then, 1 mL of this inoculum was transferred into each vial containing 1 mL of diluted product, resulting in the following test dilutions of the product, per row: 4, 8, 16, 32, 64, 128, 256, 512, 1024, 2048, 4096, and 8192. Vials containing 1 mL of M-H Broth only, without product, and 1 mL of inoculum were used as positive growth controls. Wells containing 1 mL of diluted product (a two-fold dilution series) and 1 mL of M-H Broth without any of the bacterial isolates were used as negative controls. Vials were incubated at 35 ± 1 °C for 21 ± 3 h. After incubation, the lowest concentration (the highest dilution) of the product that completely inhibits visible growth was recorded; the minimum inhibitory concentration (MIC). To check for purity after inoculation of the vials, bacterial suspensions made in saline were streaked onto Columbia agar with 5% sheep blood agar. Following overnight incubation at 37 °C, cultures were checked for morphologically characteristic colonies.

### 4.6. Antibiotic Resistance Test

The 8 classes of antimicrobials agents (β-lactams, aminoglycosides, polymyxins, fluoroquinolones, tetracyclines, macrolides, lincosamides, and sulfonamide) as well as florfenicol, tiamulin, and tylvalosin were used for the antimicrobial susceptibility test.

Antimicrobial susceptibility was assessed by determining the MIC values using a 96 well MICRONAUT Special Plates with antimicrobials: amoxicillin and clavulanic acid (AMX/CL), amoxicillin (AMX), cefquinome (CFQ), ceftiofur (CFTI), cephalexin (CFX), cloxacillin (CLO), colistin (COL), cefapiryna (CPH), docycycline (DOX), enrofloxacin (ENR), erythromycin (ERY), florfenicol (FLR), gentamicin (GEN), lincomycin (LIN), lincomycin/specinicin (LIN/SP), nafcillin (NAF), neomycin (NEO), norfloxacin (NOR), oxytetracycline (OXY), benzylpenicillin (PG), streptomycin (STR), trimethoprim-sulfamethoxazole (TR/SMX), tiamulin (TIA), tylosin (TYL), and tylvalosin (TYLV) (MERLIN Diagnostika GmbH, Bremen, Germany) was used. The MICs were interpreted according to the Clinical and Laboratory Standards Institute (CLSI) and FDA breakpoints [47].

Simultaneously, antimicrobial susceptibility was assessed by determining the MIC values using a VITEK^®^ 2 System and AST-GN96 cards for Gram-negative bacteria (BioMérieux). The AST card is essentially a miniaturized and abbreviated version of the doubling dilution technique for MICs determined by the microdilution method [46]. With using AST-GN96 susceptibility for amoxicillin/clavulanic acid (AMX/CL); ampicillin (AM), cephalexin (CFX), cephalotin (CF), ceftriaxone (CFP), cefequinome (CFQ), ceftiofur (CFTI), enrofloxacin (ENR), florfenicol (FLR), flumequine (UB), gentamicin (GEN), imipenem (IPM), marbofloxacin (MRB), neomycin (NEO), polymyxin B (PB), tetracycline (TE), and trimethoprim/sulfamethoxazole (TR/SMX) and additional antibiotics, ampicillin (AM), cefalotin (CF), cefoperazone (CFP), imipenem (IPM), flumequine (UB), marbofloxacin (MRB), tetracycline (TE), polymyxin B (PB), and trimethoprim/sulfamethoxazole (TR/SMX) was assessed.

The isolates were subjected to antibiotic susceptibility tests against 33 antibiotics belonging to 13 different classes using MIC method Merlin MICRONAUT (MERLIN Diagnostika GmbH, Germany) and AST-GN 96 CARD and VITEK2 system (Biomerieux, France). The AST card is essentially a miniaturized and abbreviated version of the doubling dilution technique for MICs determined by the microdilution method [48]. The multiple antibiotics resistance (MAR) phenotypes were performed for isolates showing resistance to more than two antibiotics [49]. MAR index was using the formula:$$\mathrm{MAR}=\frac{\mathrm{Number}\;\mathrm{of}\;\mathrm{resistance}\;\mathrm{to}\;\mathrm{antibiotics}}{\mathrm{Total}\;\mathrm{number}\;\mathrm{of}\;\mathrm{antibiotics}\;\mathrm{tested}}$$

### 4.7. Detection of Antimicrobial Resistance Genes by Multiplex PCR

Bacterial DNA isolation was performed using an automated method (AutoPure96, China). Fourteen resistance genes (*aad*A, *str*A/*str*B, *aph*A1, *aph*A2, *aad*B, *tet*A, *tet*B, *sul*1, *sul*2, *flo*R, *bla*_TEM_, *bla*_SHV_, *bla*_CMY-2_, and *bla*_PSE-1_) were analyzed by conventional PCR using specific primer pairs in multiplex or single PCR reaction The primer sequences predicted PCR product sizes and references shown in Table 6.

## 5. Conclusions

Our results show the antibacterial effect (in vitro) of a unique mixture of phytobiotics on selected *Salmonella* strains isolated from material collected from flocks where *Salmonella* infection has been confirmed, as well as on classic reference strains used in the routine diagnosis of *Salmonella* in broiler chicken flocks. This is the first study where this unique phytobiotics mixture was used.

These promising observations are extremely important not only for poultry production, but also for the current situation in human and animal medicine related to increasing antibiotics resistance. In addition, due to the legal prohibition of treating *Salmonella* in broiler flocks in a classic way—with the use of antibiotics, the tested mixture of phytobiotics may potentially support the control and spread of this type of bacteria (as a part of non-antibiotic strategy), and ultimately reduce the incidence of infections in humans related to the consumption of poultry products. Therefore, the reduction of *Salmonella* from the farm seems to be crucial to contribute to food safety. What is also important is that the use of phytobiotics as an antibiotics alternative (also based on our results) seems to be essential in protecting the therapeutic effect of antibiotics and indirectly reducing (preventing) the growing antibiotics resistance.

In summary, our pilot observations on the in vitro model provide valuable information on new potentially effective solutions for the control of *Salmonella* in poultry production. Due to the fact that these are in vitro observations, the next step in assessing the effectiveness of this unique mixture of phytobiotics should include an in vivo model experiment, taking into account not only the potential antibacterial activity, but also (due to the rich composition of the mixture and the multidirectional action of phytobiotics) on production parameters and meat quality.

## Figures and Tables

**Figure 1 antibiotics-11-00868-f001:**
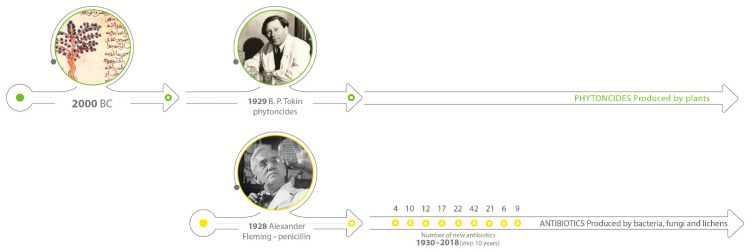
Timeline of antibacterial natural and synthetic compounds development.

**Figure 2 antibiotics-11-00868-f002:**
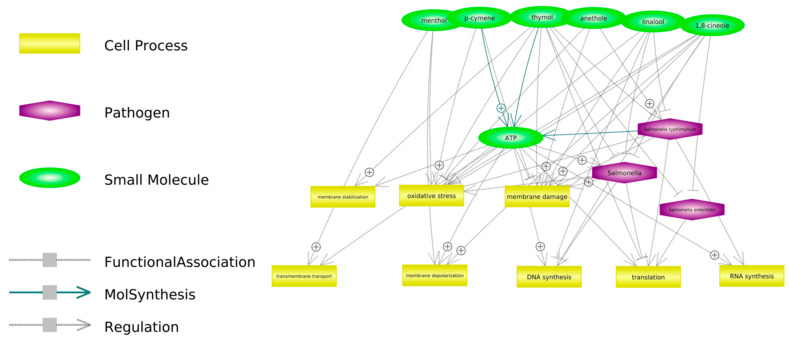
Potential antibacterial activity of selected components of phytobiotics mixture used in the study. The links were created in the Pathway Studio Web (USA) software on the basis of available publications.

**Table 1 antibiotics-11-00868-t001:** Overview of antimicrobial resistance (AMR) of *Salmonella* transmission pathways originating from poultry production (based on [2]).

Country	AMR Transmission Pathway(s)	Findings
Kenya	Indirect transmission to backyard poultry.	*Salmonella* spp. were isolated and detected the presence of class 1 integrons beta-lactamase genes from backyard chicken feces.
Vietnam	Intensive chicken farming Occupational exposure.	Demonstrated an association with AMR *Salmonella* spp. in farmers and intensively farmed poultry.
EU	Zoonotic.	Human and food-production animals had a moderate to high prevalence of *E. coli* and *Salmonella* resistant to ampicillin, tetracyclines, and sulfonamides, a high to extremely high resistance to fluoroquinolones in *Salmonella* spp., *E. coli,* and *Campylobacter* recovered from humans, broilers, fattening turkeys, and poultry carcasses/meat, and low levels of bacteria resistant to colistin in food-producing animals. Multiple drug resistance (MDR) *Salmonella enterica* serotype Infantis recovered from broilers.
USA	Zoonotic.	Moderate levels of *Salmonella* resistant to ciprofloxacin associated to direct and indirect contact with animal feces. MDR *Salmonella enterica* serotype Infantis recovered from broiler’s meat. Whole-genome sequencing revealed that this strain was identified from sick people returning from South America, and it is rapidly spreading among people and animal populations.

**Table 2 antibiotics-11-00868-t002:** Number of notifications counted for each combination of hazard/product category/notifying country.

Hazard	Product Category	Notifying Country	Notification
*Salmonella*	Poultry meat and poultry meat products.	Poland	70
*Salmonella*	Poultry meat and poultry meat products.	Lithuania	63
*Salmonella*	Poultry meat and poultry meat products.	France	50
*Salmonella*	Poultry meat and poultry meat products.	Italy	44

**Table 3 antibiotics-11-00868-t003:** MIC of analyzed mixture.

*Salmonella* Strains	Sample Source	Dilution											
		1:2	1:4	1:16	1:32	1:64	1:118	1:256	1:512	1:1024	1:2048	1:4096	1:8192
*Salmonella* Typhimurium (BO4)	Boot swabs	−	−	−	−	−	−	−	+	+	+	+	+
*Salmonella* Typhimurium (BO4)	Boot swabs	−	−	−	−	−	−	−	−	+	+	+	+
*Salmonella* Typhimurium (BO4)	Boot swabs	−	−	−	−	−	−	−	+	+	+	+	+
*Salmonella* Typhimurium (BO4)	Boot swabs	−	−	−	−	−	−	−	−	+	+	+	+
*Salmonella* Kentucky (CO8)	Boot swabs	−	−	−	−	−	−	−	+	+	+	+	+
*Salmonella* Kentucky (CO8)	Dust	−	−	−	−	−	−	−	−	+	+	+	+
*Salmonella* Enteritidis (DO9)	Intestines	−	−	−	−	−	−	−	+	+	+	+	+
*Salmonella* Enteritidis (DO9)	Liver	−	−	−	−	−	−	−	−	−	+	+	+
*Salmonella* Enteritidis (DO9)	Spleen	−	−	−	−	−	−	−	−	+	+	+	+
*Salmonella* Enteritidis (DO9)	Boot swabs	−	−	−	−	−	−	−	−	+	+	+	+
*Salmonella* Enteritidis (DO9)	Boot swabs	−	−	−	−	−	−	−	−	+	+	+	+
*Salmonella* Enteritidis (DO9)	Boot swabs	−	−	−	−	−	−	−	+	+	+	+	+

“−“—dilution was not effective agains analysed bacteria, “+”—dilution was effective agains analysed bacteria.

**Table 4 antibiotics-11-00868-t004:** Multiple antibiotic resistance index (MAR index) and phenotype pattern of *Salmonella enterica* spp. *Enterica,* all identified serovars isolates from samples of poultry.

*Salmonella* Strains	Sample Source	Phenotypic AntimicrobialResistance Profile	MAR Index
*Salmonella* Typhimurium (BO4)	Boot swabs	CFX-CPH-GEN-NEO-STR	0.2
	Boot swabs	AMP-AMX-CFX-CFT-CPH-CFT-CFP-GEN-STR-ENR-UB-MRB-FLR-LIN/SP	0.56
	Boot swabs	CFX-CPH-GEN-NEO-STR-ENR-MRB-FLR-LIN/SP	0.36
	Boot swabs	CFX-CPH-GEN-NEO-STR	0.2
*Salmonella* Kentucky (CO8)	Boot swabs	AMP-AMX-AMX/CL-CFX-CFT-CPH-CFP-GEN-NEO-STR-ENR-UB-MRB-NOR-DOX-OXY-TET-LIN/SP	0.72
	Dust	AMP-AMX-CFX-CFT-CPH-CFP-GEN-STR-ENR-UB-MRB-DOX- OXY-TET	0.56
*Salmonella* Enteritidis (DO9)	Intestines	CPH-GEN-STR-DOX-OXY-TET	0.24
	Liver	GEN-STR-UB-LIN/SP	0.16
	Spleen	CPH-GEN-NEO-STR-UB	0.2
	Boot swabs	CPH-GEN-STR-UB	0.16
	Boot swabs	CFX-CPH-GEN-NEO-STR-UB	0.24
	Boot swabs	CPH-GEN-STR-LIN/SP	0.16

Letter abbreviations correspond to the individual antibiotics according to list: ampicilln (AMP), amoxicillin (AMX), amoxicillin and clavulanic acid (AMX/CL), cephalexin (CFX), cefalotin (CFT), cefapirin (CPH), cefoperazone (CFP), gentamicin (GEN), neomycin (NEO), streptomycin (STR), enrofloxacin (ENR), flumequine (UB), marbofloxacin (MRB), norfloxacin (NOR), docycycline (DOX), oxytetracycline (OXY), tetracycline (TET), florfenicol (FLR), and lincomycin/specinicin (LIN/SP).

**Table 5 antibiotics-11-00868-t005:** Distribution of resistance genes amongst *Salmonella* isolates.

*Salmonella* Strains	Sample Source	Resistance Phenotype	Resistance Genes
*Salmonella* Typhimurium (BO4)	Boot swabs	CFX-CPH-GEN-NEO-STR	*aad*A, *str*A/*str*B, *aph*A1, *aph*A2
	Boot swabs	AMP-AMX-CFX-CFT-CPH-CFP-CFTI-GEN-STR-ENR-UB-MRB-FLR-LIN/SP	*bla*_CMY-2,_*bla*_PSE-1,_*bla*_TEM,_*aad*A, *str*A/*str*B, *flo*R
	Boot swabs	CFX-CPH-GEN-NEO-STR-ENR-MRB-FLR-LIN/SP	*aad*A, *str*A/*str*B, *aph*A1, *aph*A2, *flo*R
	Boot swabs	CFX-CPH-GEN-NEO-STR	*aad*A, *str*A/*str*B, *aph*A1, *aph*A2
*Salmonella* Kentucky (CO8)	Boot swabs	AMP-AMX-AMX/CL-CFX-CFT-CPH-CFP-GEN-NEO-STR-ENR-UB-MRB-NOR-DOX-OXY-TET-LIN/SP	*bla*_CMY-2,_*bla*_PSE-1,_*bla*_TEM,_*aad*A, *str*A/*str*B, *aph*A1, *aph*A2, *tet*A, *tet*B
	Dust	AMP-AMX-CFX-CFT-CPH-CFP-GEN-STR-ENR-UB-MRB-DOX-OXY-TET-LIN/SP	*bla*_CMY-2,_*bla*_PSE-1,_*bla*_TEM,_*aad*A, *str*A/*str*B, *tet*A, *tet*B
*Salmonella* Enteritidis (DO9)	Intestines	CPH-GEN-STR-DOX-OXY-TET	*aad*A, *str*A/*str*B, *tet*A, *tet*B
	Liver	GEN-STR-UB-LIN/SP	*aad*A, *str*A/*str*B
	Spleen	CPH-GEN-NEO-STR-UB	*aad*A, *str*A/*str*B, *aph*A1, *aph*A2
	Boot swabs	CPH-GEN-STR-UB	*aad*A, *str*A/*str*B
	Boot swabs	CFX-CPH-GEN-NEO-STR-UB	*aad*A, *str*A/*str*B, *aph*A1, *aph*A2
	Boot swabs	CPH-GEN-STR-LIN/SP	*aad*A, *str*A/*str*B

**Table 6 antibiotics-11-00868-t006:** Primers sequences for detection of antimicrobial resistance genes in the *Salmonella* spp. isolate and multiplex PCR annealing temperature (based on [50]).

Multiplex PCR	Gene/Antibiotic	Primer Sequences 5′–3′	Annealing Temperature	Product Size (bp)
**Multiplex** 1	*aad*A streptomycin	F-GTG GAT GGC GGC CTG AAG CC R-AAT GCC CAG TCG GCA GCG	63 °C	525 bp
**Multiplex** 1	*str*A/*str*B streptomycin	F-ATG GTG GAC CCT AAA ACT CT R-CGT CTA GGA TCG AGA CAA AG	63 °C	893 bp
**Multiplex** 2	*aph*A1 neomycin	F-ATG GGC TCG CGA TAA TGT C R-CTC ACC GAG GCA GTT CCA T	55 °C	634 bp
**Multiplex** 2	*aph*A2 neomycin	F-GAT TGA ACA AGA TGG ATT GCR-CCA TGA TGG ATA CTT TCT CG	55 °C	347 bp
**Multiplex** 2	*aad*B gentamicin	F-GAG GAG TTG GAC TATGGA TT R-CTT CAT CGG CAT AGT AAA AG	55 °C	208 bp
**Multiplex** 3	*tet*A tetracycline	F-GGC GGT CTT CTT CAT CAT GC R-CGG CAG GCA GAG CAA GTA GA	63 °C	502 bp
**Multiplex** 3	*tet*B tetracycline	F-CGC CCA GTG CTG TTG TTG TC R-CGC GTT GAG AAG CTG AGG TG	63 °C	173 bp
**Multiplex** 4	*sul*1 sulfamethoxazole	F-CGG CGT GGG CTA CCT GAA CG R-GCC GAT CGC GTG AAG TTC CG	66 °C	433 bp
**Multiplex** 4	*sul*2 sulfamethoxazole	F-CGG CAT CGT CAA CAT AAC CT R-TGT GCG GAT GAA GTC AGC TC	66 °C	721 bp
**Single PCR**	*flo*R *florfenicol*	F-CACGTTGAGCCTCTATATGG R-ATGCAGAAGTAGAACGCGAC	61 °C	888 bp
**Multiplex** 5	*bla*_TEM_ ampicillin	F-TTAACTGGCGAACTACTTAC R-GTCTATTTCGTTCATCCATA	55 °C	247 bp
**Multiplex** 5	*bla*_SHV_ ceftiofur	F-AGGATTGACTGCCTTTTTG R-ATTTGCTGATTTCGCTCG	55 °C	393 bp
**Multiplex** 5	*bla*_CMY-2_ ceftiofur	F-GACAGCCTCTTTCTCCACA R-TGGACACGAAGGCTACGTA	55 °C	1000 bp
**Single PCR**	*bla*_PSE-1_ ampicillin	F-GCAAGTAGGGCAGGCAATCA R-GAGCTAGATAGATGCTCACAA	60 °C	461 bp

Abbreviations: bp—base pairs.

## Data Availability

Not applicable.

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
