# Peer review of "In Vitro Evaluation of Antimicrobial Effect of Phytobiotics Mixture on Salmonella spp. Isolated from Chicken Broiler"

_antibiotics, 2022, doi:10.3390/antibiotics11070868_

Round 1

Reviewer 2 Report

The article "In vitro evaluation of antimicrobial effect of phytobiotics mix-2 ture on Salmonella spp. isolated from chicken broiler" is a very interesting topic and contains good data which merits publication.The introduction and conclusion sections are well organized and step by step self-explained, very easy for readers to understand the aim of the article. But before its final decision, I would like to ask for minor suggestions.

1. First of all kindly change all the names of microbes into Italic throughout the paper.

2. There are some typo mistakes and some English language errors, kindly also carefully revise the paper line by line from start till end.

 detailed comments for the article. 1. Why the authors didn't use positive control for their studies? 2. What was the quantity of the phytoncides mixtures (Gm/mg)?? during doing anti-bacterial tests? 3. For the supplementary data can the authors provide the MIC and MBC pictures?    

Reviewer 3 Report

I have evaluated the manuscript (Antibiotics-1783214) titled “In vitro evaluation of antimicrobial effect of phytobiotics mixture on Salmonella spp. isolated from chicken broiler” by IwiÅ„ski and co-workers, and the author has discussed the antibacterial activity of phytoncides mixture in vitro against Salmonella spp. isolated from broiler farms. Excellent presentation of results in the manuscript and clearly describing the outcome, however, the author could concise the discussion to avoid repetition of discussion of the same subject.  I found the document interesting for the readers and follow the scope of the journal Antibiotics.

I would like to recommend the article could be published in Antibiotics, with minor revision.

The authors could make the following minor changes.

1.     The author could have used an abbreviation with the full name when it first appears in the manuscript.

2.     Introduction should be short, concise, and to the point. No need to discuss and elaborate well know topics.

3.     All the tables should include footnotes.

4.     The author could have discussed the composition of the mixtures.

5.     Resolution of the figures is poor; the author needs to change it.

6.     Some parts of the conclusion could move to the discussion section.

 7.     Supplementary documents should include title, author’s name, and table of content. All the tables in the supporting documents should contain footnotes. Recheck all the tables as some columns are not aligned with the entries.

8.     The author could include the following relevant references:

 (a)   Aljumaah, M. R., Alkhulaifi, M. M., & Abudabos, A. M. (2020). In vitro Antibacterial Efficacy of Non-Antibiotic Growth Promoters in Poultry Industry. The journal of poultry science57(1), 45–54. https://doi.org/10.2141/jpsa.0190042  

(b)  Adhikari B, Hernandez-Patlan D, Solis-Cruz B, Kwon YM, Arreguin MA, Latorre JD, Hernandez-Velasco X, Hargis BM and Tellez-Isaias G (2019) Evaluation of the Antimicrobial and Anti-inflammatory Properties of Bacillus-DFM (Norum™) in Broiler Chickens Infected With Salmonella Enteritidis. Front. Vet. Sci. 6:282. doi: 10.3389/fvets.2019.00282
